# Enhanced biofilm and extracellular matrix production by chronic carriage versus acute isolates of *Salmonella* Typhi

Aishwarya Devaraj[1], Juan F. González[1,2,3], Bradley Eichar[1,2], Gatan Thilliez[4], Robert A. Kingsley[4,5], Stephen Baker[6,7], Marc W. Allard[8], Lauren O. Bakaletz[1,3,9], John S. Gunn[1,2,3,9,10]*, Steven D. Goodman[1,3,9,10]*

1 Center for Microbial Pathogenesis, Research Institute at Nationwide Children's Hospital, Columbus, Ohio, United States of America, 2 Department of Microbial Infection and Immunity, The Ohio State University, Columbus, Ohio, United States of America, 3 Department of Pediatrics, College of Medicine, The Ohio State University, Columbus, Ohio, United States of America, 4 Quadram Institute Bioscience, Norwich, United Kingdom, 5 University of East Anglia, Norwich, United Kingdom, 6 Cambridge Institute of Therapeutic Immunology and Infectious Disease, University of Cambridge School of Clinical Medicine, Cambridge Biomedical Campus, Cambridge, United Kingdom, 7 Department of Medicine, University of Cambridge School of Clinical Medicine, Cambridge Biomedical Campus, Cambridge, United Kingdom, 8 Food and Drug Administration-FDA, College Park, Maryland, United States of America, 9 Infectious Diseases Institute, The Ohio State University, Columbus, Ohio, United States of America, 10 Oral and GI Microbiology Research Affinity Group, Nationwide Children's Hospital, Columbus, Ohio, United States of America

* John.Gunn@NationwideChildrens.org (JSG); Steven.Goodman@NationwideChildrens.org (SDG)

**Data Availability Statement:** All relevant data are within the manuscript and its Supporting Information files.

## Abstract

*Salmonella* Typhi is the primary causative agent of typhoid fever; an acute systemic infection that leads to chronic carriage in 3–5% of individuals. Chronic carriers are asymptomatic, difficult to treat and serve as reservoirs for typhoid outbreaks. Understanding the factors that contribute to chronic carriage is key to development of novel therapies to effectively resolve typhoid fever. Herein, although we observed no distinct clustering of chronic carriage isolates via phylogenetic analysis, we demonstrated that chronic isolates were phenotypically distinct from acute infection isolates. Chronic carriage isolates formed significantly thicker biofilms with greater biomass that correlated with significantly higher relative levels of extracellular DNA (eDNA) and DNABII proteins than biofilms formed by acute infection isolates. Importantly, extracellular DNABII proteins include integration host factor (IHF) and histone-like protein (HU) that are critical to the structural integrity of bacterial biofilms. In this study, we demonstrated that the biofilm formed by a chronic carriage isolate *in vitro*, was susceptible to disruption by a specific antibody against DNABII proteins, a successful first step in the development of a therapeutic to resolve chronic carriage.

## Author summary

*Salmonella* Typhi, a human restricted pathogen is the primary etiologic agent of typhoid fever, an acute systemic infection that has a global incidence of 21 million cases annually. Although the acute infection is resolved by antibiotics, 3–5% of individuals develop

**Funding:** This work was supported by NIH grant R01DC011818 to SDG and LOB; NIH grant R01 AI116917 and start-up funds from Abigail Wexner Research Institute at Nationwide Children's Hospital to JSG; Wellcome senior research fellowship 215515/Z/19/Z to SB; BBSRC Institute Strategic Programme Microbes in the Food Chain BB/R012504/1 and its constituent projects BBS/E/F/000PR10348 and BBS/E/F/000PR10349 to RAK. The funders did not play any role in the study design, data collection and analysis, decision to publish, or preparation of the manuscript.

**Competing interests:** The authors have declared that no competing interests exist.

chronic carriage that is difficult to resolve with antibiotics. A majority of these indivuals serve as reservoirs for further spread of the disease. Understanding the differences between acute and chronic carrier strains is key to design novel targeted approaches to undermine carriage. Here, we demonstrated that chronic carrier strains although not genotypically distinct from acute strains, formed thicker biofilms with greater relative levels of extracellular eDNA and DNABII proteins than those formed by acute infection isolates. We also demonstrated that an antibody against DNABII proteins significantly disrupted biofilms formed by a chronic carrier strain and therefore supported development of therapeutic use of this antibody to attenuate chronic carriage.

## Introduction

*Salmonella enterica* is a facultative intracellular, Gram-negative gammaproteobacterium, which is classified into six subspecies that are further subtyped into more than 2000 serovars or serotypes based on the expression of surface antigens [1,2]. In humans, *S. enterica* serovars cause non-typhoidal [3] and typhoidal [4] illness that results in significant morbidity and mortality worldwide. Non-typhoidal *S. enterica* causes gastroenteritis with a global burden of 93 million cases and 155,000 deaths annually [5]. *Salmonella enterica* serovar Typhi (*S.* Typhi) is a human-restricted pathogen and the primary etiologic agent of typhoid fever with an incidence of 21 million cases each year that results in 200,000 deaths annually [6].

Enteric or typhoid fever is an acute systemic infection that is commonly caused by poor sanitation and the consumption of contaminated food or water [7]. Although acute infections are resolved by treatment with antibiotics, a high incidence of antibiotic resistance with about 60% of strains that exhibit multidrug resistance, exacerbates the morbidity and mortality, particularly, within typhoid endemic regions [8]. Most importantly, 3–5% of the individuals with an acute infection ultimately develop chronic asymptomatic carriage in the gallbladder [9]. This chronic carriage state not only serves as a reservoir for further spread of the disease via bacterial shedding in feces, but is also difficult to resolve with antibiotics [10–13]. Given the grave public health concern of both acute infections and chronic carriage of *S.* Typhi, it is important to understand the development of each of these states to design and test targeted approaches to resolve the more recalcitrant chronic carriage.

The hallmark of chronic carriage of *S.* Typhi is the successful colonization of the gallbladder and biofilm formation on the surface of gallstones [14,15]. Biofilms are organized three-dimensional multicellular communities encased in self-produced extracellular polymeric substances (EPS) that is comprised of polysaccharides, extracellular DNA [eDNA], proteins and lipids [16,17]. Biofilm formation on the surface of gallstones is thought to protect the resident bacteria from the harsh environment within the gallbladder (e.g. presence of bile), host immune effectors, and antibiotics [18]. Although the negative impact of the chronic carriage state on human health has been recognized for over a century, very little is known about the genotypic and phenotypic differences between acute infection and chronic carriage isolates of *S.* Typhi.

In order to better characterize the differences in phenotypic and genotypic characteristics of acute infection and chronic carriage isolates of *S.* Typhi, in this study, we sequenced a lab strain *S.* Typhi Ty2 (JSG698), and multiple acute infection (14 isolates from patients with an acute infection) and chronic carriage isolates (6 isolates from chronic carriers). We also examined the ability of each of these isolates to form biofilms *in vitro* (as biofilm formation is a hallmark of chronic carriage) and compared the relative levels of multiple EPS components

(eDNA, DNABII proteins and lipopolysaccharides [19,20]) within their respective biofilms to determine if these correlated with differences in the magnitude of the biofilms formed. Perhaps most importantly, we have also shown that *S.* Typhi biofilms, similar to those formed by 22 single and multi-species bacterial biofilms, incorporate DNABII proteins that stabilize the eDNA lattice-like structure [21–24]. Sequestration of DNABII proteins from bacterial biofilm via specific antibodies, results in collapse of the EPS which causes release of biofilm-resident bacteria that are highly sensitive to antibiotics and immune factors [21,22,25–30].

In this study, we demonstrated that chronic carriage isolates formed thicker biofilms than acute infection isolates and further, that of those components tested, the increase in biofilm formation by chronic carriage isolates correlated only with greater steady state levels of eDNA and DNABII proteins within the biofilm EPS. Overall, these data suggested that during carriage, chronic *S.* Typhi isolates may undergo pathoadaptation in the gallbladder for enhanced biofilm capabilities. Indeed, we also demonstrated that we could disrupt biofilms formed by chronic carriage isolates with a specific antibody that targets the DNABII proteins which thereby supported development of therapeutic use of this antiserum to potentially attenuate the chronic carriage of *S.* Typhi.

## Results

### Molecular and genetic typing of acute and chronic *S.* Typhi strains

To infer the phylogenetic relationship of acute and chronic *S.* Typhi isolates from Ohio, Mexico City, and Vietnam (Table 1), we determined the whole-genome sequence using short read sequence technology. A total of 1,777 single nucleotide polymorphisms in the core genome with reference to *S.* Typhi Ty2 (AE014613) were used to construct a maximum-likelihood phylogenetic tree (**Fig 1A**) in the context of 1909 global *S.* Typhi isolates reported previously [31]. The study isolates were widely distributed within the global diversity of *S.* Typhi and ten isolates were of the H58 haplotype that constitute the currently dominant global pandemic clade [32]. Clinical isolates displayed high genetic similarity with multiple clades formed (**Fig 1B**), with pairwise distance ranging from 1 to 548 SNPs in the core genome (**Fig 1C**). However, there was no apparent clustering of the sequenced isolates by the acute *versus* chronic infection status of the patient.

### Chronic carrier isolates of *S.* Typhi formed thicker biofilms than acute infection isolates

To identify any phenotypic alterations between *S.* Typhi strains isolated from acute and chronic infections, we first set out to determine the differences in biofilms formed *in vitro* by each of these strains. This is relevant as chronic carriage of *S.* Typhi is associated with biofilm formation on the surface of gallstones [14,15]. We first characterized the differences in growth of each of the *S.* Typhi strains and observed that the chronic carriage isolates (JSG3983 and JSG3984) had a significantly longer doubling time than *S.* Typhi Ty2 strain JSG698 (**S1 Fig and Table 2**). Of the these two strains only JSG3984 eventually matched the extent of growth of *S.* Typhi Ty2 strain. Due to these two growth defects, JSG3983 was therefore the only strain eliminated from our comparative analysis. Next, we allowed various *S.* Typhi chronic carriage and acute infection isolates and the lab strains *S.* Typhi Ty2 (JSG698 and JSG4383) (**Table 1**) to form biofilms *in vitro*. Since the lab strain *S.* Typhi Ty2 (JSG698) is deficient in *rpoS*, a known regulator of stationary phase gene expression, an otherwise isogenic *rpoS*+ strain (JSG4383 [33]) was also employed. First, no significant difference in biofilm formation between the *S.* Typhi Ty2 (JSG698) and the *S.* Typhi Ty2 *rpoS*+ strain (JSG4383) was observed

**Table 1. Strains used in this study.**

**Laboratory strains**

| Strain | Characteristics | | | Source |
|---|---|---|---|---|
| JSG698 | *S.* Typhi Ty2; wild-type | | | ATCC |
| JSG4383 | *S.* Typhi Ty2 *rpoS*+ | | | [35] |
| JSG1213 | *S.* Typhi Ty2 *tviB*::Kan | | | Gift of Popoff lab, Pasteur Institute [34] |
| JSG210 | *S.* Typhimurium 14028s; wild-type | | | ATCC |

**Clinical Isolates**

| Strain | Isolation source/characteristics | Infection type | Country of origin | Source |
|---|---|---|---|---|
| JSG691 (ICOPHAI17078) | Blood | Acute | USA | University of Texas Health Science Center at San Antonio |
| JSG3074 (ICOPHAI17076) | Gallstone | Chronic | Mexico | General Hospital of Mexico, Mexico City |
| JSG4123 | JSG3074 Δ*tviB* via Wanner | NA | NA | This study |
| JSG3076 (ICOPHAI17077) | Gallstone | Chronic | Mexico | General Hospital of Mexico, Mexico City |
| JSG3395 (ICOPHAI17081) | Blood | Acute | USA | Ohio Department of Health |
| JSG3400 (ICOPHAI17086) | Bile | Acute | USA | Ohio Department of Health |
| JSG3407 (ICOPHAI17082) | Stool | Acute | USA | Ohio Department of Health |
| JSG3418 (ICOPHAI17085) | Stool | Acute | USA | Ohio Department of Health |
| JSG3419 (ICOPHAI17083) | Blood | Acute | USA | Ohio Department of Health |
| JSG3431 (ICOPHAI17084) | Stool | Acute | USA | Ohio Department of Health |
| JSG3433 (ICOPHAI17079) | Blood | Acute | USA | Ohio Department of Health |
| JSG3441 (ICOPHAI17080) | Stool | Acute | USA | Ohio Department of Health |
| JSG3979 (GB169) | Gallbladder | Chronic | Vietnam | Gift of S. Baker |
| JSG3980 (GB281) | Gallbladder | Chronic | Vietnam | Gift of S. Baker |
| JSG3981 (GB31) | Gallbladder | Chronic | Vietnam | Gift of S. Baker |
| JSG3982 (GB335) | Gallbladder | Chronic | Vietnam | Gift of S. Baker |
| JSG3983 (GB266) | Gallbladder | Chronic | Vietnam | Gift of S. Baker |
| JSG3984 (GB26) | Gallbladder | Chronic | Vietnam | Gift of S. Baker |
| JSG3985 (TY421) | Unspecified | Acute | Vietnam | Gift of S. Baker |
| JSG3986 (TY312) | Unspecified | Acute | Vietnam | Gift of S. Baker |
| JSG3987 (TY311) | Unspecified | Acute | Vietnam | Gift of S. Baker |
| JSG3988 (TY102) | Unspecified | Acute | Vietnam | Gift of S. Baker |
| JSG3989 (TY261) | Unspecified | Acute | Vietnam | Gift of S. Baker |
| JSG3990 (TY96) | Unspecified | Acute | Vietnam | Gift of S. Baker |

NA = Not applicable.

as evidenced by biofilm average thickness (**Fig 2A**) and biofilm biomass (**Fig 2B**). This result suggested that the lack of *rpoS* had no significant effect on biofilm formation by the *S.* Typhi Ty2 strain JSG698.

As shown in Fig 2, when grossly assessed, while 5 out of the 6 chronic carriage isolates (83%) formed significantly thicker biofilms as compared to the lab strain *S.* Typhi Ty2 (JSG698), only 2 out of the 14 acute infection isolates (14%) formed significantly thicker biofilms as compared to lab strain. Next, we determined the average thickness and biomass of biofilms formed by chronic (indicated by the red dotted line labeled AC) versus and acute infection isolates (indicated by the blue dotted line labeled AA). In general, chronic carriage isolates formed significantly thicker biofilms ($P<0.01$) with greater biomass ($P<0.01$) compared to those formed by acute infection isolates (**Fig 2**). To further validate our confocal microscopy-based analyses, we enumerated the bacteria within each biofilm (adherent state) and demonstrated that in biofilms formed by chronic carriage isolates, on average 52.5% of the

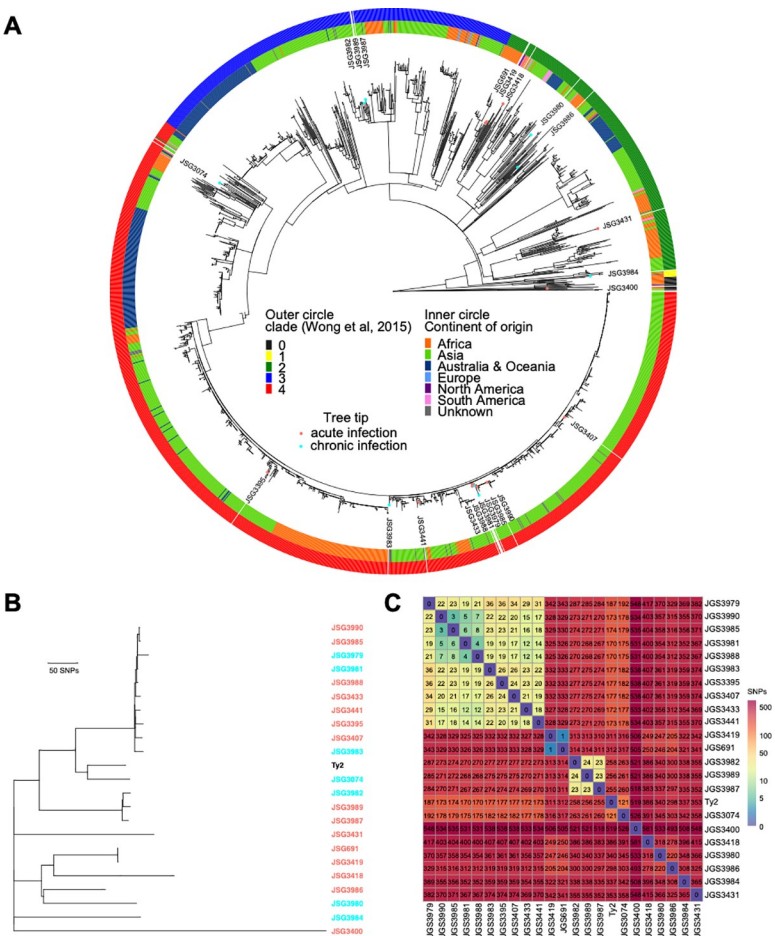

**Fig 1. Phylogenetic relationship of *S.* Typhi isolates.** (**A**) Maximum likelihood phylogenetic tree based on core genome sequence variation. This shows the relationship of *S.* Typhi strains used in this study in the context of a global collection of *S.* Typhi strains reported in (31). (**B**) Maximum likelihood phylogenetic tree showing the relationship of *S.* Typhi clinical isolates and *S.* Typhi Ty2 based on core genome sequence variation. (**C**) Single nucleotide polymorphism distance matrix and heatmap of isolates in this study. There was no apparent clustering of the sequenced isolates by the acute *versus* chronic infection status of the patient.

total bacteria were in the biofilm state, whereas this value was 26.8% for biofilms formed by acute infection isolates (**S2 Fig**). These data were consistent with our analysis derived via confocal microscopy and collectively suggested that chronic carriage isolates were significantly different from acute infection isolates in their relative ability form biofilms, *in vitro*.

## Chronic carriage isolates yielded higher steady state levels of eDNA and DNABII proteins within their biofilm EPS compared to those formed by acute infection isolates

Given that chronic carriage isolates formed a thicker biofilm with greater biomass as compared to those formed by acute infection isolates, we examined the steady state levels of various EPS components within *S.* Typhi biofilms to correlate with the differences in biofilm architecture observed between chronic carriage and acute infection isolates. *S.* Typhi whole cells (WC; extracellular) or cell lysates (Ly) were blotted on PVDF membranes and probed with either α-Vi-antigen (**S3A Fig**) or α-O9 O-antigen of the LPS (**S3B Fig**) antibodies. As shown in S4 Fig,

**Table 2. Doubling time in hours for each of the indicated *S*. Typhi strains.**

| Strain | Doubling time (hours) |
|---|---|
| *S*. Typhi Ty2 (JSG698) | 1.19 ± 0.01 |
| JSG3395 | 1.14 ± 0.01 |
| JSG3407 | 1.21 ± 0.01 |
| JSG3418 | 1.20 ± 0.01 |
| JSG3419 | 1.20 ± 0.01 |
| JSG3431 | 1.23 ± 0.02 |
| JSG3433 | 1.16 ± 0.02 |
| JSG3441 | 1.17 ± 0.01 |
| JSG3985 | 1.21 ± 0.004 |
| JSG3986 | 1.24 ± 0.01 |
| JSG3987 | 1.26 ± 0.01 |
| JSG3988 | 1.24 ± 0.004 |
| JSG3989 | 1.15 ± 0.01 |
| JSG3990 | 1.20 ± 0.01 |
| JSG3074 | 1.21 ± 0.005 |
| JSG3979 | 1.20 ± 0.02 |
| JSG3980 | 1.25 ± 0.02 |
| JSG3981 | 1.19 ± 0.01 |
| JSG3982 | 1.23 ± 0.01 |
| JSG3983 | 1.36 ± 0.02 **** |
| JSG3984 | 1.33 ± 0.02 **** |

Statistical significance of doubling times for each of the chronic carriage or acute infection strain versus the *S*. Typhi Ty2 strain, JSG698 was assessed by a one-way ANOVA followed by Dunnett's multiple comparison test.
****$P < 0.0001$.

while Vi-antigen was variably expressed by both chronic carriage and acute infection isolates, no characteristic pattern of expression was observed for either group (**S3A Fig**). *S*. Typhimurium strain 14028s and *S*. Typhi *tviB*::*Kan* [34] served as negative controls. O9 antigen was below the level of detection in WC blots of multiple chronic carriage and acute infection isolates. Additionally, no apparent difference in the expression of O9 antigen was observed between chronic carriage and acute infection isolates (**S3B Fig**). *S*. Typhimurium strain 14028s served as a negative control in these latter blots. These data suggested that chronic carriage isolates had no discernible pattern of differences from acute infection isolates with respect to the expression of either O9 antigen or Vi-antigen.

Next, we quantified the relative extracellular levels of eDNA and DNABII proteins within the biofilms formed by the indicated *S*. Typhi strains. We allowed biofilms to be formed by each strain for 40 hours, then incubated the biofilms with α-dsDNA monoclonal antibody and α-IHF$_{Ec}$ (recognizes both IHF and HU isolated from a large crossection of bacterial species; also the primary amino acid sequences of IHF and HU from *E. coli* and *Salmonella* are identical) to visualize both eDNA and DNABII proteins within the biofilm EPS by immunofluorescence microscopy (IF). The bacterial membrane stain, FM 4–64 was used to designate the overall biofilm architecture. The distribution of eDNA (teal) and DNABII proteins (purple) within the biofilm EPS of representative chronic carriage (JSG3980) and acute infection (JSG3986) isolates compared to lab strain *S*. Typhi Ty2 (JSG698) is shown in **Fig 3A**. The fluorescence intensity of eDNA (**Fig 3B**) and DNABII proteins (**Fig 3D**) were quantified, and the average fluorescence intensity of eDNA and DNABII proteins within the EPS of biofilms

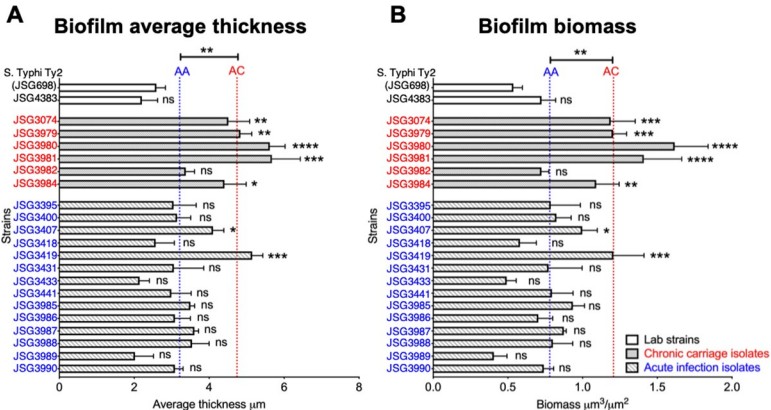

**Fig 2. Chronic carriage isolates formed thicker biofilms than acute isolates.** Biofilms of each of the indicated *S.* Typhi strains were established in a 8-well chambered coverglass slide for 40 hours. Biofilms were stained with LIVE/DEAD stain and visualized via CLSM. Images were analyzed by COMSTAT to calculate average thickness (**A**) and biomass (**B**). Bars represent the standard error of the mean (SEM). The mean of the average thickness or biomass of the chronic carriage isolates was represented by the red dotted line labeled AC and the mean of the average thickness or biomass of the acute isolates was represented by the blue dotted line labeled AA. Statistical significance of average thickness or biomass of each of the strains versus the the lab strain, *S.* Typhi Ty2 (JSG698), was assessed by a one-way ANOVA followed by Dunnett's multiple comparison test. Statistical significance between AC and AA were assessed by a one-way ANOVA followed by Tukey's multiple comparison test. *P<0.05, **P<0.01, ***P<0.001, ****P<0.0001. On average, chronic carriage isolates formed a thicker biofilm with more biomass than acute isolates. Chronic carriage isolates are indicated in red and acute infection isolates are indicated in blue.

formed by acute infection and chronic carriage isolates was represented by the blue and red dotted lines labeled AA and AC respectively. As shown in Fig 3B and 3D, on average, chronic carriage isolates had significantly higher steady state levels of both eDNA (*P*<0.01) and DNA-BII proteins (*P*<0.01) within their biofilm EPS as compared to acute infection isolates. However, despite the greater absolute abundance of eDNA and DNABII proteins within biofilms formed by chronic carriage isolates, the relative abundance of eDNA and DNABII proteins [as determined by the ratio of fluorescence intensity of eDNA or DNABII proteins to the fluorescence intensity of bacteria] was not different (see dotted lines AC and AA in **Fig 3C and 3E**).

Thus, regardless of biofilm size, the underlined ratio of eDNA and DNABII to the number of cells in the biofilm remains constant. This result suggested that in order for biofilms to incorporate bacterial cells they are rate limited by constant ratios of eDNA and DNABII i.e. the more eDNA and DNABII present within the biofilms, the more bacteria they can incorporate. Collectively, these data suggested that while chronic carriage and acute infection isolates did not have a discernible difference in expression of either LPS or Vi-antigen, the relative amount of eDNA and DNABII proteins were significantly greater within biofilms formed by chronic carriage isolates than that within the biofilms formed by acute infection isolates due to the presence of more bacteria.

### Biofilms formed by *S.* Typhi were disrupted upon incubation with a DNABII-specific antibody

DNABII proteins serve as linchpin proteins to stabilize the eDNA lattice structure and in turn, sequestration of DNABII proteins from the biofilm EPS with specific antibodies results in significant disruption of single and multi-species biofilms *in vitro*, *ex vivo* and *in vivo* [21,22,24–27,29,30,35–37]. Accordingly, we hypothesized that the greater the steady state levels of DNA-BII proteins present within a biofilm, the greater the dose of DNABII-directed antibodies that would be required for biofilm disruption. Since biofilms formed by chronic carriage isolates

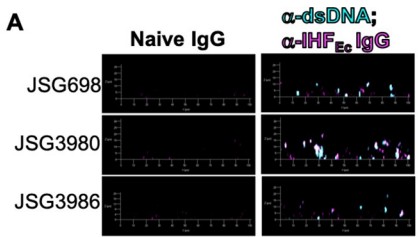

**I. Quantification of eDNA within *S.* Typhi biofilms**

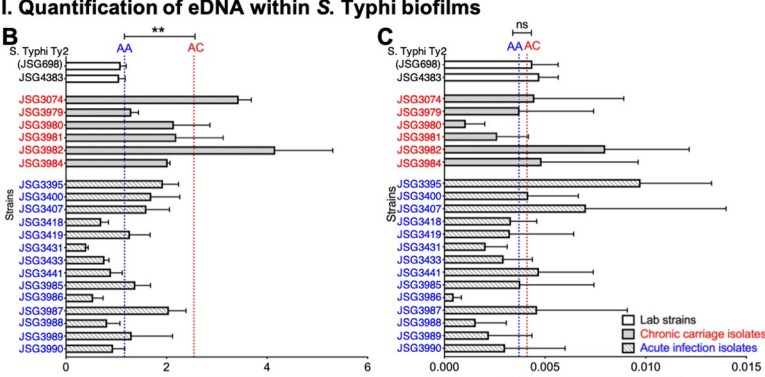

**II. Quantification of extracellular DNABII proteins within *S.* Typhi biofilms**

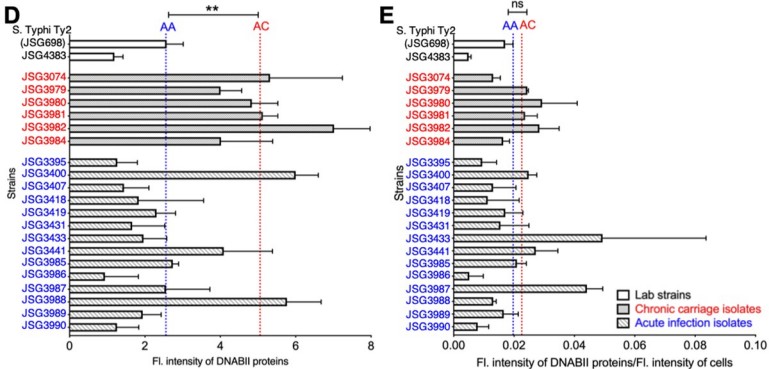

**Fig 3. Chronic carriage isolates have higher steady levels of extracellular DNA and DNABII proteins within their biofilm EPS than acute isolates.** Biofilms of each of the indicated *S.* Typhi strains were established in an 8-well chambered coverglass slide for 40 hours. Biofilms were labeled with α-dsDNA monoclonal antibody, α-IHF$_{Ec}$ and FM 4–64 and visualized via CLSM. (**A**) Representative immunofluorescence images of a lab wild type strain *S.* Typhi Ty2 (JSG698), chronic carriage isolate (JSG3980) and acute isolate (JSG3986). eDNA is visualized in teal and DNABII proteins in purple. Images were analyzed by ImageJ to quantify the fluorescence intensity of eDNA (**B**), DNABII proteins (**D**), and bacteria. Fluorescence intensity of eDNA and DNABII proteins were normalized to cells and plotted in (**C**) and (**E**), respectively. Bars represent the SEM. AA and AC represent the averages of acute infection isolates and chronic carriage isolates, respectively. Statistical significance between AC and AA were assessed by a one-way ANOVA followed by Tukey's multiple comparison test. **P<0.01. On average, chronic carriage isolates had higher steady state levels of both eDNA and DNABII proteins within the biofilm EPS as compared to acute isolates. Chronic carriage isolates are indicated in red and acute infection isolates are indicated in blue.

had a greater amount of DNABII proteins, here we wanted to determine if these could nonetheless be disrupted with specific antibodies. To this end, we allowed the lab strain *S.* Typhi Ty2 (JSG698) and the chronic carriage isolate (JSG3074; used here as a representative bacterium) to form biofilms for 24 hours, then incubated these biofilms with α-IHF$_{Ec}$, for 16 hours. Naive IgG was used as a negative control. As shown in **Fig 4**, biofilms formed by both the lab strain and the chronic carriage isolate were significantly disrupted by α-IHF$_{Ec}$ (significant decrease in biomass as compared to naive IgG), although as hypothesized, a greater

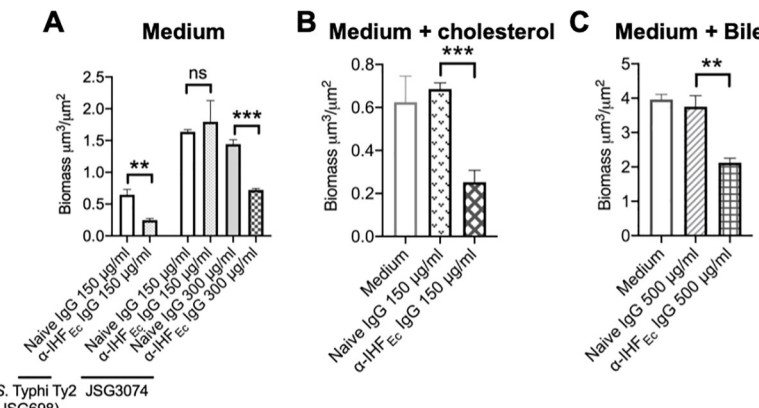

**Fig 4. DNABII-specific antibody disrupted biofilms formed by chronic carriage isolate of S. Typhi.** (**A**) Biofilms formed by lab wild type strain *S.* Typhi Ty2 (JSG698) or chronic carriage isolate (JSG3074) or (**B, C**) JSG698 were established in an 8-well chambered coverglass slide in TSB (A), chambered coverglass slide coated with 5 mg/ml cholesterol (**B**), or TSB+0.5% ox bile extract (**C**) for 24 hours. Biofilms were incubated with medium alone, naive IgG, or α-IHF$_{Ec}$ IgG at the indicated concentration for 16 hours. Biofilms were stained with LIVE/DEAD stain and visualized via CLSM. Images were analyzed by COMSTAT to calculate biomass. Bars represent the SEM. **P<0.01, ***P<0.001 via unpaired *t* test. Biofilms formed by *S.* Typhi Ty2 (JSG698) in the presence of cholesterol and bile, as well as a chronic carriage isolate of *S.* Typhi, were significantly disrupted by DNABII-specific antibody (α-IHF$_{Ec}$).

concentration of α-IHF$_{Ec}$ was required to achieve a similar effect in the biofilm formed by the chronic carriage isolate as that observed for the lab strain. This requirement for a greater concentration of α-IHF$_{Ec}$ for biofilm disruption was consistent with the presence of higher steady state levels of DNABII proteins within biofilm EPS of the chronic carriage isolate as compared to the lab strain (**Fig 3**).

Since *S.* Typhi attaches to gallstones (that primarily consists of cholesterol) and encounters bile in the gallbladder during establishment of a chronic carriage state, and that bile induces biofilm formation of S. Typhi [15], we also determined the efficacy of biofilm disruption by α-IHF$_{Ec}$ in the presence of cholesterol or bile. Biofilms formed by *S.* Typhi Ty2 strain were disrupted by α-IHF$_{Ec}$ as observed by a significant reduction in biofilm biomass as compared to naive IgG in the presence of either cholesterol (chambered coverglass slide was coated with cholesterol to mimic the surface of gallstones) or 0.5% bile (**Fig 4B and 4C**). Although bile significantly increased the biomass within the biofilms formed by the lab strain with a concomitant increase in the expression of DNABII proteins both intracellularly (**S4A Fig**) as well as that incorporated within the biofilm EPS (**S4B and S4C Fig**), a greater concentration of α-IHF$_{Ec}$ (500 μg/ml) nevertheless disrupted biofilms formed under conditions designed to mimic those within the gallbladder. Collectively, these results suggested that the biofilms formed by chronic carriage isolates were susceptible to disruption by α-IHF$_{Ec}$ *in vitro* (albeit at a higher α-IHF$_{Ec}$ concentration), an essential first step to develop these antibodies as a therapeutic that can potentially resolve chronic carriage of *S.* Typhi.

## Discussion

The chronic carriage state of *S.* Typhi is difficult to treat and these individuals also serve as reservoirs that can cause community outbreaks of typhoid fever. A more comprehensive characterization of the differences between acute infection and chronic carriage isolates is critical in order to understand the development of the chronic carriage state and design novel, specific targeted approaches to undermine carriage.

In this study, we first characterized the genotypic and phenotypic differences between acute infection and chronic carriage isolates of S. Typhi and demonstrated that each of these strains were widely distributed within the global diversity of S. Typhi strains with no evident clustering. Although large-scale rearrangements between ribosomal RNA operons have been documented in longitudinal isolates of an S. Typhi strain from a chronic carrier [38], our sequence analysis of multiple acute infection and chronic carriage isolates from various geographical locations suggested no distinct clustering of strains based on the chronic carrier versus acute infection status of the patient. Previous genome sequence comparison analysis of longitudinal isolates of persistent S. Typhimurium revealed several SNPs in global virulence regulatory genes such as *rpoS*, *dksA*, *melR* etc., that can cause pleiotropic changes in the transcription of genes during persistence [39]. Also, acquisition of mobile genetic elements has been documented during persistence that is directly attributed to the antibiotic resistant phenotype of chronic carriage isolates [38]. Thus, while in our study clustering of the acute infection versus chronic isolates does not occur, genetic events do occur in the gallbladder environment that likely affect strain phenotypes.

In contrast, we demonstrated that on average, chronic carriage isolates formed thicker biofilms than acute isolates. This result is in line with previous findings which suggest that the chronic carriage state of S. Typhi is facilitated by formation of biofilms on gallstones within the gallbladder wherein S. Typhi predominantly persists [14,15]. Biofilm resident bacteria are highly recalcitrant to antibiotic therapy, and clearance by host immune effectors (reviewed in [40]), which in turn, facilitates chronic carriage. Much of this resistance is associated with the EPS components within biofilms, which in addition to maintenance of the biofilm structure, also provides a physical barrier to and directly compromises the activities of host immune effectors [16]. eDNA is a key structural component of bacterial biofilms [41–44] and is also incorporated within S. Typhimurium biofilms wherein it facilitates resistance to antibiotics and antimicrobial peptides [45]. In this study, we have shown that the thicker biofilms of chronic carriage isolates correlated with a greater relative level of eDNA than acute infection isolates that could also likely contribute to their recalcitrance to treatment modalities.

Previous studies have identified several S. Typhi uniquely expressed antigens from chronic carrier sera that include membrane proteins, lipoproteins and hemolysin-related proteins [46]. Additionally, persistent/chronic carriage isolates of S. Typhi exhibit increased expression of iron transporters and other factors that provide resistance to host antimicrobial peptides [47]. We have now identified DNABII proteins to be differentially and highly expressed within the EPS of the biofilms formed by chronic carriage isolates of S. Typhi. Whereas the role of this increased steady state level of eDNA and DNABII proteins, beyond their contribution to biofilm structural integrity is not yet known, a DNABII member expressed by Mycobacteria is known to possess ferroxidase activity. It has been suggested that ferrous iron (used for fenton chemistry to generate peroxide as a host defense) could be de-toxified by released DNABII protein-mediated oxidation to the ferric state, suitable for bacterial iron transport and utilization [48]. In addition, environmental factors such as bile within the gallbladder upregulate the expression of EPS components [49,50] that are required for biofilm formation on gallstones [20,51,52]. In line with these findings, herein we also observed that bile upregulated the intracellular expression of DNABII proteins which corresponded with a concomitant increased incorporation of these proteins into the biofilm EPS. Furthermore, the DNABII protein IHF positively regulates the expression of curli, another EPS component that contributes to biofilm formation by S. Typhi [53]. Collectively, our new data contribute to the findings of others to suggest that the microenvironment wherein S. Typhi persists results in positive regulation of components (proteins, polysaccharides etc.,) that promote the chronic carriage state.

The DNABII family of proteins are highly conserved and one of the members of the family, HU is ubiquitously expressed by eubacteria. DNABII proteins bind to and bend DNA [54,55] and thereby play a critical role in the intracellular nucleoid structure and function [56]. These proteins are also found in the EPS of multiple single and multi-species bacterial biofilms [21–24,27,30,35,57]. In this study, we now expanded upon our previous observations to demonstrate that extracellular DNABII proteins were incorporated within the EPS of biofilms formed by both acute infection and chronic carriage isolates of *S.* Typhi and that these proteins are critical to the structural integrity of biofilms formed by *S.* Typhi. These data are in line with our previous findings that DNABII proteins serve as linchpin proteins to stabilize the eDNA lattice structure and their sequestration, via incubation with a specific antibody, disrupts multiple bacterial biofilms *in vitro* [21,22,24,26,27,29,36], resolves bacterial biofilms *in vivo* [27,30,37], and dissolves sputum solids *ex vivo* [35]. Moreover, pre-incubation with the aforementioned DNABII-specific antibody reduces binding of uropathogenic *E. coli* to cultured bladder epithelial cells [58] and also, reduces survival of *Burkholderia cenocepacia* that have been phagocytized by murine cystic fibrosis macrophages [22]. Collectively, these results indicate that the extracellular presence of the DNABII family of proteins is involved in facets of pathogenesis in addition to its structural role in the EPS. Finally, here for the first time, we demonstrated that for multiple isolates of *S.* Typhi, there was a constant ratio between cells, eDNA and DNABII regardless of biofilm size. This latter finding strongly implied that eDNA and DNABII proteins, and their associated specific ratios, were rate limiting for biofilm formation and further suggested that more eDNA and DNABII within the biofilm matrix can shift the partitioning of cells from the planktonic state into the adherent biofilm state. Should this paradigm exist throughout eubacterial biofilms, it would indicate that the most likely rationale is for there to be a uniform common eDNA-DNABII-dependent EPS that exists within the biofilms formed by diverse species. Given that the DNABII proteins are universally conserved and have been well documented to stabilize bacterial biofilm structure, the ability to significantly disrupt a biofilm formed by a chronic carriage isolate with antibody directed against a DNABII protein is an exciting first step in the development of a novel therapeutic to attenuate chronic carriage of *S.* Typhi.

## Materials and methods

### Bacterial strains and growth conditions

Bacterial strains used in this study are listed in Table 1. All clinical *S.* Typhi isolates tested positive by Vi-antigen agglutination test before use. Strains were streaked on lysogeny broth (LB) agar plates and incubated at 37˚C for 18–20 hours. Single colonies were used to start overnight (O/N) liquid cultures. Planktonic cells were grown at 37˚C on a rotating drum in LB or tryptic soy broth (TSB). Antibiotics, when needed, were used at the following concentrations: chloramphenicol, 25 μg/ml; ampicillin, 50 or 100 μg/ml, kanamycin, 45 μg/ml; and streptomycin, 100 μg/ml. The deletion of *tviB* in strain JSG3074 was generated by the λ-Red mutagenesis method [59] with the following primers: upstream primer JG2934 (5'-ataaaattttagtaaaggattaa-taagagtgttcggtatagtgtaggctggagctgcttc– 3'), downstream primer JG2935 (5'–gtccgtagttcttcg-taagccgtcatgattacaatctcaccatatgaatatcctccttag– 3'), and verified with primers JG2936 (5'-tcagcgacttctgttctattcaagtaagaaaggggtacgg– 3'), and JG2937 (5' -gctcctcactgacggacgtgc-gaacgtcgtctagattatg- 3'). Antibiotic resistance markers were swapped out using pCP20 [60]. The mutant was verified via sequencing.

### Molecular and genetic typing of acute and chronic *S.* Typhi strains

Clinical *S.* Typhi strains acquired from Vietnam had been previously whole-genome sequenced before use in this study. All remaining isolates were whole-genome sequenced as

part of a consortium with the U.S. Food and Drug Administration (FDA). Genomic DNA of each strain was isolated from overnight cultures using DNeasy Blood and Tissue kit (Qiagen, CA, United States). Isolates were sequenced using Illumina's MiSeq platform (Illumina, Inc., CA, United States). Sample preparation and the sequencing library was prepared using the Nextera XT Sample Preparation Kit and then sequenced for $2 \times 250$ cycles. The assembled sequences were annotated using the NCBI Prokaryotic Genomes Annotation Pipeline (PGAP) and have been deposited at DDBJ/EMBL/GenBank. Genomes are publicly available from Pathogen Detection at NCBI (search ICOPHAI IDs at https://www.ncbi.nlm.nih.gov/pathogens/isolates/).

The maximum-likelihood phylogenetic tree was constructed from the core SNP alignment of all isolates using snippy version 3.0 as previously described [61]. *S.* Typhi Ty 2 (JSG698) was used as a reference for the mapping and SNP calling steps. *S.* Paratyphi A (strain A270) and *S.* Typhimurium 14028s (JSG210) where included as outgroups to infer the true root for the tree in Fig 1A and 1B respectively, but were removed for mapping and variant calling and the resulting tree manually rooted. RAxML (version 8.2.10) [62] was used to construct maximum likelihood phylogenetic trees from the core SNP alignment, with the generalized time-reversible model and a Gamma distribution to model site-specific rate variation (the GTR+Γ substitution model; GTRGAMMA in RAxML). Support for the maximum-likelihood phylogeny was assessed via 400 rapid bootstraps based on the MRE_IGN-based Bootstrapping criterion.

## Visualization and quantification of biofilms formed by various *S.* Typhi strains

*S.* Typhi strains were cultured on TSB agar for 18–20 h at 37˚C, 5% $CO_2$ in a humidified atmosphere, then suspended in TSB to an OD of 0.65 at 490 nm. Cultures were diluted 1:6 in TSB, then incubated statically at 37˚C, 5% $CO_2$ until an OD of 0.6 was reached at 490 nm. The cultures were diluted 1:2500 in TSB and 200 μl of this suspension was inoculated into each well of an eight-well chambered cover glass slide (Fisher Scientific). Slides were incubated statically for 16 h at 37˚C, 5% $CO_2$ in a humidified atmosphere at which time, spent medium was aspirated and replaced with fresh TSB. After an additional 8 h (24 h total incubation time), spent medium was aspirated and replaced with fresh TSB. After an additional 16 h (40 h total incubation time), biofilms were stained with LIVE/DEAD *Bac*Light Bacterial Viability kit for microscopy (Molecular Probes) as per manufacturer's instructions, then fixed (1.6% paraformaldehyde- 0.025% glutaraldehyde, 4% acetic acid in 0.2M phosphate buffer, pH 7.4). Biofilms were visualized via Zeiss 510 Meta-laser scanning confocal microscope and imaged with a x63 objective. Biofilm average thickness and biomass were determined by COMSTAT analysis [63]. The assay was repeated three times on separate days. Data are presented as mean ± SEM.

## Determination of the number of bacteria in the planktonic or biofilm state

Biofilms were established as described in 'Visualization and quantification of biofilms' for 40 h. Bacteria within the adherent biofilm as well as those within the culture fluid above the biofilm (e.g. planktonic state) were enumerated as described previously [57].

## Dot blot assay

Bacterial cultures were grown overnight in TSB at 37˚C with aeration, fixed in 4% paraformaldehyde (PFA), and normalized to OD600 = 0.8 in PBS. Normalized cultures were then divided into a lysate group and a non-lysate group (no further processing). Lysate group was boiled for 10 min to lyse the cells. Bacterial dilutions (1:6 for Vi-antigen; 1:20 and 1:50 for O9 antigen) were prepared, and 200 μl was spotted onto methanol-activated PVDF membranes using a

suction manifold device. The blots were dried and blocked at 4˚C O/N with 5% milk buffer, followed by incubation in either α-Vi-antigen (1:1000) or α–O9 antigen (1:1000). The blots were washed 3 times, 15 minutes each, in tris-buffered saline with polysorbate 20 (TBST) and incubated with α-mouse IgG 1:2000 (for O9 antigen) or α- rabbit IgG 1:2000 (for Vi Antigen) antibodies conjugated with horseradish peroxidase (Bio- Rad) O/N at 4˚C prior to visualization using the Bio-Rad Chemi-Doc XRS system.

## Visualization and quantification of eDNA and DNABII proteins within the biofilm EPS of various *S*. Typhi strains

Biofilms were established as described in 'Visualization and quantification of biofilms' for 40 h. Unfixed biofilms were washed once with sterile PBS and incubated with either: 5 µg/ml mouse isotype control IgG 2a; 5 µg/ml α-dsDNA monoclonal antibody; 7.5 µg/ml naive rabbit IgG; or 7.5 µg/ml α-IHF$_{Ec}$ in 5% BSA in PBS for 1 h at room temperature. Biofilms were washed once with PBS and incubated with 1:200 dilution of each of goat α-mouse IgG 405, goat α-rabbit IgG 488 and FM4-64 for 1 h at room temperature. Biofilms were washed once with PBS then imaged with a x63 objective on a Zeiss 800 Meta-laser scanning confocal microscope (Zeiss). Three-dimensional images were reconstructed with AxioVision Rel 4.8 (Zeiss). Fluorescence intensity of DNABII proteins, eDNA and bacteria were quantified by ImageJ software and relative abundance of eDNA or DNABII proteins were determined by the ratio of fluorescence intensity of eDNA or DNABII proteins to the fluorescence intensity of bacteria. The assay was repeated three times on separate days. Bars represent the mean ± SEM.

## Disruption of *S*. Typhi biofilms with α-IHF$_{EC}$

*S*. Typhi strains were cultured on TSB agar for 18–20 h at 37˚C, 5% $CO_2$ in a humidified atmosphere, then suspended in TSB to an OD of 0.65 at 490 nm. Cultures were diluted 1:6 in TSB then incubated statically at 37˚C, 5% $CO_2$ until an OD of 0.6 was reached at 490 nm. The cultures were diluted1:2500 in TSB or TSB supplemented with ox bile extract or human bile to 0.5%, and 200 µl of this suspension was inoculated into each well of an eight-well chambered cover glass slide. Slides were coated with cholesterol as described [50]. Slides were incubated statically for 16 h at 37˚C, 5% $CO_2$ in a humidified atmosphere at which time, spent medium was aspirated and replaced with fresh TSB. After an additional 8 h (24 h total incubation time), the spent medium was aspirated and replaced with fresh TSB or TSB that contained either naive IgG or α-IHF$_{Ec}$ IgG (150 µg/ml– 500 µg/ml). Slides were incubated statically for 16 h at 37˚C, 5% $CO_2$ in a humidified atmosphere. Biofilms were stained and imaged as described in 'Visualization and quantification of biofilms'.

## Supporting information

**S1 Fig. Characterization of planktonic growth of various *S*. Typhi strains.** Exponential phase cultures of each of the indicated strains was diluted 1:1000 in TSB and incubated at 37˚C with continuous shaking. Absorbance at 490 nm were measured every 15 min for 16 hours. (**A**) Growth curve plot of chronic carriage isolates versus the lab wild type strain *S*. Typhi Ty2 (JSG698), in triplicates. (**B**) Growth curve plot of acute isolates versus the *S*. Typhi Ty2 strain (JSG698), in triplicate. JSG3983 and JSG3984 were the only strains wherein a defect growth rate was observed. JSG3983 also did not grow to the same extent as *S*. Typhi Ty2 strain (JSG698).
(TIF)

**S2 Fig. Enumeration of bacteria within biofilms formed by various *S*. Typhi strains.** Biofilms of each of the indicated *S*. Typhi strains were established on a 8-well chambered coverglass slide for 40 hours. Biofilms were washed twice in sterile PBS, suspended in PBS, and enumerated on TSB agar. Of the total bacteria within the chambered coverglass, those that were adherent within the biofilm were enumerated and values presented as percent of total. Bars represent the SEM. The mean of the percent biofilm bacteria of the chronic carriage isolates was represented by the dotted line labeled AC and the mean of the percent biofilm bacteria of the acute infection isolates was represented by the dotted line labeled AA. Statistical significance of average thickness or biomass of each of the strains versus the *S*. Typhi Ty2 strain, JSG698 was assessed by a one-way ANOVA followed by Dunnett's multiple comparison test. Statistical significance between AC and AA were assessed by a one-way ANOVA followed by Tukey's multiple comparison test. $^{*}P<0.05$, $^{**}P<0.01$, $^{***}P<0.001$, $^{****}P<0.0001$. On average, chronic carriage isolates had more bacteria within their biofilms than acute isolates. Chronic carriage isolates are indicated in red and acute infection isolates are indicated in blue. (TIF)

**S3 Fig. Expression profile of Vi antigen and O9 antigen in various *S*. Typhi strains.** Representative images of dot blots for expression of Vi antigen (**A**) and LPS (**B**). Either whole cells (WC) or cell lysates (Ly) of each of the indicated strains were spotted on methanol-activated PVDF membranes using a suction manifold device. The blots were probed with α-Vi antigen antibody (**A**) or α- O9 antigen antibody (**B**). No clear differences in the expression of Vi antigen or O9 antigen were observed between chronic carriage isolates and acute isolates. (TIF)

**S4 Fig. Bile increased the expression of DNABII proteins intracellularly and increased the incorporation of DNABII proteins extracellularly within the biofilm EPS.** (**A**) Planktonically grown *S*. Typhi Ty2 (JSG698) was pelleted, lysed and proteins were resolved on an SDS-PAGE gel. DNABII proteins were quantified by Western blot. (**B**) *S*. Typhi Ty2 (JSG698) biofilms were formed *in vitro* in the presence of TSB or TSB + 0.5% ox bile extract for 40 hours. Biofilms were labeled with α-IHF$_{Ec}$ and FM 4–64 and visualized via CLSM. Representative immunofluorescence images that show the distribution of DNABII proteins within the biofilms. Images were analyzed by ImageJ to quantify the fluorescence intensity of DNABII proteins and bacteria. Fluorescence intensity of DNABII proteins (**C**) and fluorescence intensity of DNABII proteins normalized to cells (**D**) were plotted. Bars represent the SEM. $^{*}P<0.05$, $^{**}P<0.01$ via unpaired t test. Bile increased the intracellular and extracellular steady state levels of DNABII proteins within *S*. Typhi biofilms. (TIF)

## Acknowledgments

We thank Wondwossen Gebreyes for his help with genomic sequencing, Kristen Reeve for her early studies on the biofilm phenotypes of the acute and chronic isolates and John Buzzo for his comments.

## Author Contributions

**Conceptualization:** Aishwarya Devaraj, Steven D. Goodman.

**Data curation:** Aishwarya Devaraj, Juan F. González, Bradley Eichar, Robert A. Kingsley, Stephen Baker, Marc W. Allard.

**Formal analysis:** Aishwarya Devaraj, Juan F. González, Bradley Eichar, Gatan Thilliez, Robert A. Kingsley.

**Funding acquisition:** Lauren O. Bakaletz, John S. Gunn, Steven D. Goodman.

**Investigation:** Aishwarya Devaraj, Juan F. González, Bradley Eichar.

**Methodology:** Aishwarya Devaraj, Bradley Eichar, Gatan Thilliez, Robert A. Kingsley.

**Project administration:** Robert A. Kingsley, Stephen Baker, Marc W. Allard, Lauren O. Bakaletz, John S. Gunn, Steven D. Goodman.

**Resources:** Robert A. Kingsley, Stephen Baker, Lauren O. Bakaletz, John S. Gunn, Steven D. Goodman.

**Software:** Gatan Thilliez, Robert A. Kingsley.

**Supervision:** Robert A. Kingsley, Stephen Baker, Marc W. Allard, Lauren O. Bakaletz, John S. Gunn, Steven D. Goodman.

**Validation:** Aishwarya Devaraj.

**Visualization:** Aishwarya Devaraj, Juan F. González, Gatan Thilliez, Robert A. Kingsley.

**Writing – original draft:** Aishwarya Devaraj, Gatan Thilliez, Robert A. Kingsley, Marc W. Allard.

**Writing – review & editing:** Juan F. González, Robert A. Kingsley, Lauren O. Bakaletz, John S. Gunn, Steven D. Goodman.

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
