## [Decision Letter · Decision Letter 0]

9 Nov 2020

Dear Dr. Goodman,

Thank you very much for submitting your manuscript "Enhanced Biofilm and Extracellular Matrix Production by Chronic Carriage versus Acute Isolates of Salmonella Typhi" for consideration at PLOS Pathogens. As with all papers reviewed by the journal, your manuscript was reviewed by members of the editorial board and by several independent reviewers. The reviewers appreciated the attention to an important topic. Based on the reviews, we are likely to accept this manuscript for publication, providing that you modify the manuscript according to the review recommendations.

Sincerely,

Andreas J Baumler

Associate Editor

PLOS Pathogens

Renée Tsolis

Section Editor

PLOS Pathogens

Kasturi Haldar

Editor-in-Chief

PLOS Pathogens

orcid.org/0000-0001-5065-158X

Michael Malim

Editor-in-Chief

PLOS Pathogens

orcid.org/0000-0002-7699-2064

Reviewer Comments (if any, and for reference):

Reviewer's Responses to Questions

**Part I - Summary**

Reviewer #1: Devaraj et al. characterize a series of acute and chronic Salmonella Typhi strains, showing that, independent of lineage, the chronic strains tend to make “better” biofilms, consistent with their ability or selection to colonize gall stones. The experiments are generally straight forward, although the results are correlative without any molecular explanation. The paper is more wordy than necessary and could be shortened significantly. My other comments also focus on presentation.

Reviewer #2: In this study, Devaraj et al characterize a small collection of S. Typhi isolates to investigate whether strains isolated from chronic or acute typhoid fever exhibit genomic or phenotypic differences that could point towards pathoadaptation. Based on the genome sequence of these isolates, the authors generate a phylogenetic tree; this analysis fails to identify obvious patterns that correlate with disease state (chronic vs acute). The authors then analyze biofilm formation of each strain in vitro. Strains isolated from chronically-infected patients tended to form thicker biofilm and these biofilms contained more extracellular DNA and DNABII proteins. In contrast, no difference in the production of the Vi antigen were noted. Antibodies directed against S. Typhi DNABII proteins disrupted the biofilm and decreased biofilm-associated biomass.

Overall, the experiments are well-executed, the manuscript is well-written and easy to follow. All major conclusions are supported by the data. The finding that S. Typhi may adapt during chronic infections to produce more biofilm and thus ensure better gall bladder colonization is novel and exciting. Previous studies have reported effects of anti-DNABII on bacterial biofilms and as such, this concept and technology is not entirely novel. However, this was not known for S. Typhi and is new information. I only have some minor comments regarding the writing and data presentation.

**Part II – Major Issues: Key Experiments Required for Acceptance**

Reviewer #1: None

Reviewer #2: None

**Part III – Minor Issues: Editorial and Data Presentation Modifications**

Reviewer #1: 1. Define DNABII proteins in the abstract

2. Both the introduction and discussion could be shortened significantly without loss of important content.

3. It is not clear that the pulse-field gel experiment adds anything to the paper given the subsequent genome sequencing.

4. Line 198. It would be better to call and label JS698 Ty2. You do this in some figures, but not all. JSG4383 isn’t in the strain table so I can’t see the genotype. I wonder if (line 200) you mean that you repaired or replaced the mutant rpoS allele with wt rather than “complemented” it in trans.

5. Line 233. You don’t really mean that “LPS was absent” versus lack of antigen. Rephrase. Indeed, what do you mean by anti-LPS antibody? If it is anti-O9, for example, you just say so.

6. Line 244. “…also the translated proteins of IHF and HU…” is somewhat awkward.

7. Fig 1. Barely visible in Fig 1A, you demarcate acute and chronic strains with very small red and blue dots and this color scheme is carried through to 2B. You need to make this distinction more obvious by repeating the mark in 2B (larger dots would be nice) or saying it in the legend. …and why not carry it through to 2C?

8. Supp Fig 2. Growths curves need to be plotted with the OD on a log scale.

Minor comments:

9. Line 58 individuals

10. Line 60. …chronic carrier state, although not genotypically distinct…

11. Line 119. …immune effectors, and antibiotics…

12. Line 121. …that negatively affect its…

13. Line 124. “…that alter the transcriptional landscape of chronic carriage of Salmonella…” This doesn’t make sense as written.

Reviewer #2: Fig. S2: Bacterial growth curve are easier to interpret when plotted on semi-log plot (log of OD, time in linear scale). The exponential growth phase is then clearly identifiable as a linear feature on this type of plot. The doubling time, based on exponential growth, should be calculated and reported. Also, is the growth defect referring to a slower growth rate during exponential phase or a lower recovery (density) in stationary phase? Based on the current plot, it certainly is the latter, but the former is difficult to assess.

I recommend that standard bacterial genetics nomenclature should be followed throughout the manuscript. It is a bit unusual to see that wild-type strains are referred to by a number assigned by the investigator. JSG698 and JSG210 should be denoted as Ty2 and 14028S in the manuscript (e.g. line 201 and Fig. 2). Also, if the tviB mutant JSG1213 was previously published by the Popoff group, please provide this reference (maybe PMID: 8574397?).

PLOS authors have the option to publish the peer review history of their article (what does this mean?). If published, this will include your full peer review and any attached files.

Reviewer #1: No

Reviewer #2: No
---

## [Editor Report · Decision Letter 1]

2 Dec 2020

Dear Dr. Goodman,

We are pleased to inform you that your manuscript 'Enhanced Biofilm and Extracellular Matrix Production by Chronic Carriage versus Acute Isolates of Salmonella Typhi' has been provisionally accepted for publication in PLOS Pathogens.

Best regards,

Andreas J Baumler

Associate Editor

PLOS Pathogens

Renée Tsolis

Section Editor

PLOS Pathogens

Kasturi Haldar

Editor-in-Chief

PLOS Pathogens

orcid.org/0000-0001-5065-158X

Michael Malim

Editor-in-Chief

PLOS Pathogens

orcid.org/0000-0002-7699-2064
---

## [Editor Report · Acceptance letter]

4 Jan 2021

Dear Dr. Goodman,

We are delighted to inform you that your manuscript, "Enhanced Biofilm and Extracellular Matrix Production by Chronic Carriage versus Acute Isolates of Salmonella Typhi," has been formally accepted for publication in PLOS Pathogens.

Best regards,

Kasturi Haldar

Editor-in-Chief

PLOS Pathogens

orcid.org/0000-0001-5065-158X

Michael Malim

Editor-in-Chief

PLOS Pathogens

orcid.org/0000-0002-7699-2064